# Blind Image Restoration via Fast Diffusion Inversion

**Hamadi Chihaoui**     **Abdelhak Lemkhenter**     **Paolo Favaro**
Computer Vision Group, Institute of Informatics, University of Bern, Switzerland
{hamadi.chihaoui,abdelhak.lemkhenter,paolo.favaro}@unibe.ch

## Abstract

Image Restoration (IR) methods based on a pre-trained diffusion model have demonstrated state-of-the-art performance. However, they have two fundamental limitations: 1) they often assume that the degradation operator is completely known and 2) they alter the diffusion sampling process, which may result in restored images that do not lie onto the data manifold. To address these issues, we propose Blind Image Restoration via fast Diffusion inversion (BIRD) a blind IR method that jointly optimizes for the degradation model parameters and the restored image. To ensure that the restored images lie onto the data manifold, we propose a novel sampling technique on a pre-trained diffusion model. A key idea in our method is not to modify the reverse sampling, *i.e.*, not to alter all the intermediate latents, once an initial noise is sampled. This is ultimately equivalent to casting the IR task as an optimization problem in the space of the input noise. Moreover, to mitigate the computational cost associated with inverting a fully unrolled diffusion model, we leverage the inherent capability of these models to skip ahead in the forward diffusion process using large time steps. We experimentally validate BIRD on several image restoration tasks and show that it achieves state of the art performance. Project page: https://hamadichihaoui.github.io/BIRD.

## 1   Introduction

Recent advances in generative learning due to the development of diffusion models [7, 18] have led to models capable of generating detailed and realistic high-resolution images. In addition to their data generation capabilities, these models also provide an implicit representation of the distribution of the data they train on, which can be used for other applications, such as image restoration (IR). Indeed, several approaches have emerged to solve inverse problems using pre-trained diffusion models [8, 11, 21]. Those approaches start from a random noise vector as the diffusion input and introduce a projection after each diffusion reverse step to enforce the consistency with the corrupted image. As shown by Chung [3] this procedure alters the original diffusion sampling process, and may cause the generated image to leave the data manifold during the iterative denoising process, which ultimately results in unrealistic image samples. Moreover, most of these methods are non-blind as they assume to have full knowledge of the degradation model, which is not practical in real-world applications.

To overcome all these challenges, we propose a novel blind image restoration method that jointly optimizes the degradation model parameters and the restored image only at test time (and thus separately for each new image). We call this method BIRD, which stands for Blind Image Restoration via fast Diffusion inversion. Since BIRD does not need to pre-train a model for the degradation operator, it can be used immediately on a wide range of IR tasks such as Gaussian deblurring, motion deblurring, super-resolution and denoising (see Figure 2). In this work, we show that having a strong image prior and ensuring that the restored image never leaves the data manifold during the reconstruction process are fundamental properties to generalize to a wide range of image degradation problems. As shown in Figure 1, BIRD (see (f) to (i)) reconstructs realistic images at each iteration, while BlindDPS (see (b) to (e)) may initially reconstruct non realistic images. We observe that even

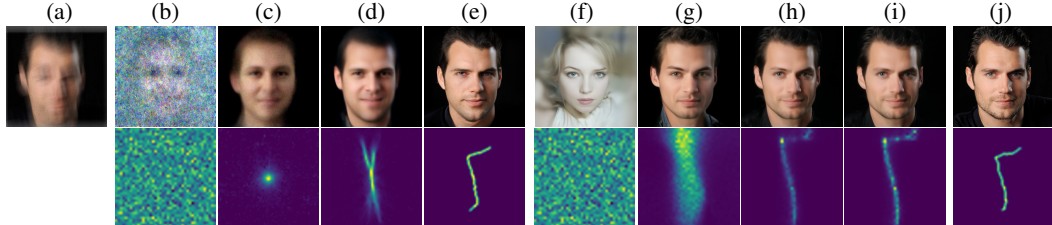

Figure 1: Blind image deblurring with unknown motion blur. (a): blurry input image. From (b) to (e): The top row shows the predictions of BlindDPS [1] as the iterations increase; the bottom row shows the corresponding estimated blur kernel. From (f) to (i): Same estimates as in (b) to (e), but obtained from BIRD, our proposed method. (j) is the ground truth sharp image (top) and ground truth blur kernel (bottom). Notice that BlindDPS [1] trains a score-based model for the kernel estimation, while BIRD does not use any training and can adapt to any new kernel directly at test time. BIRD yields always natural images at every iteration of the reconstruction procedure. Finally, notice that despite recovering a suboptimal blur kernel, the image reconstructed with BIRD is more similar to the ground truth image than with BlindDPS thanks to the robustness of our image generation procedure.

though we have a more primitive blur reconstruction procedure (we directly regress the motion blur kernel), the reconstructed image is still more similar to the ground truth one (see column (l)) than the final image estimate from BlindDPS (see (e) top), which uses a quite accurate blur kernel estimate (see (e) bottom). This speaks of the importance of defining a strong image prior first and foremost.

To define the image prior, BIRD uses a pre-trained Denoising Diffusion Implicit Model (DDIM) [17] and exploits the existing deterministic correspondence between noise and images in DDIMs by casting the inverse restoration problem as a latent estimation problem, *i.e.*, where the latent variable is the input noise to the diffusion model. In contrast to prior work, we do not alter the reverse sampling, *i.e.*, all the intermediate latents, once an initial noise is sampled. To then apply our image prior, we cast the IR task as an alternating optimization between the unknown restored image and the unknown parameters of the degradation model. However, a direct implementation of the optimization procedure in the case of a diffusion model used as a black box would be computationally demanding and ultimately impractical. To mitigate the substantial computational cost associated with inverting a fully unrolled diffusion model, we leverage, for the first time in IR tasks, the inherent capability of these models to skip ahead in the forward diffusion process using arbitrarily large time steps. We show that our method is able to achieve state of the art performance across a wide range of blind image restoration tasks. Our main contributions can be summarized as follows

1. BIRD is the first to cast an IR task as a latent optimization problem (where only the initial noise is optimized) in the context of diffusion models; our optimization procedure aims at generating images that lie on the data manifold at every iteration;

2. BIRD is computationally efficient; we propose a fast diffusion model inversion in the context of image restoration, without fine-tuning nor retraining;

3. We achieve state of the art results on CelebA and ImageNet for different blind IR tasks, such as Gaussian and motion deblurring, super-resolution, and denoising.

## 2   Related Work

**Blind and non-blind image restoration methods.**    An image restoration task is often cast as an inverse problem, where the degradation model is explicitly known. For example, in image deblurring the degradation model can be described by a convolution with a blur kernel. Methods that explicitly assume the exact knowledge of the blur kernel are called *non-blind* methods. More general methods that assume only knowledge of the type of degradation (*e.g.*, blurring), but not the values of its parameters, are instead called *blind*. Some recent examples of non-blind methods are [3] and DPS [2]. [3] points out that relying on an iterative procedure consisting of reverse diffusion steps and a projection-based consistency step runs the risk of stepping outside of the data manifold, a risk they mitigate using an additional correction term. DPS [2] proposes a more general framework to handle both the non-linear and noisy cases. DDNM [21] proposes a zero-shot framework for

| Gaussian Deblur | Motion Deblur | Denoising | 4× SR |
|---|---|---|---|

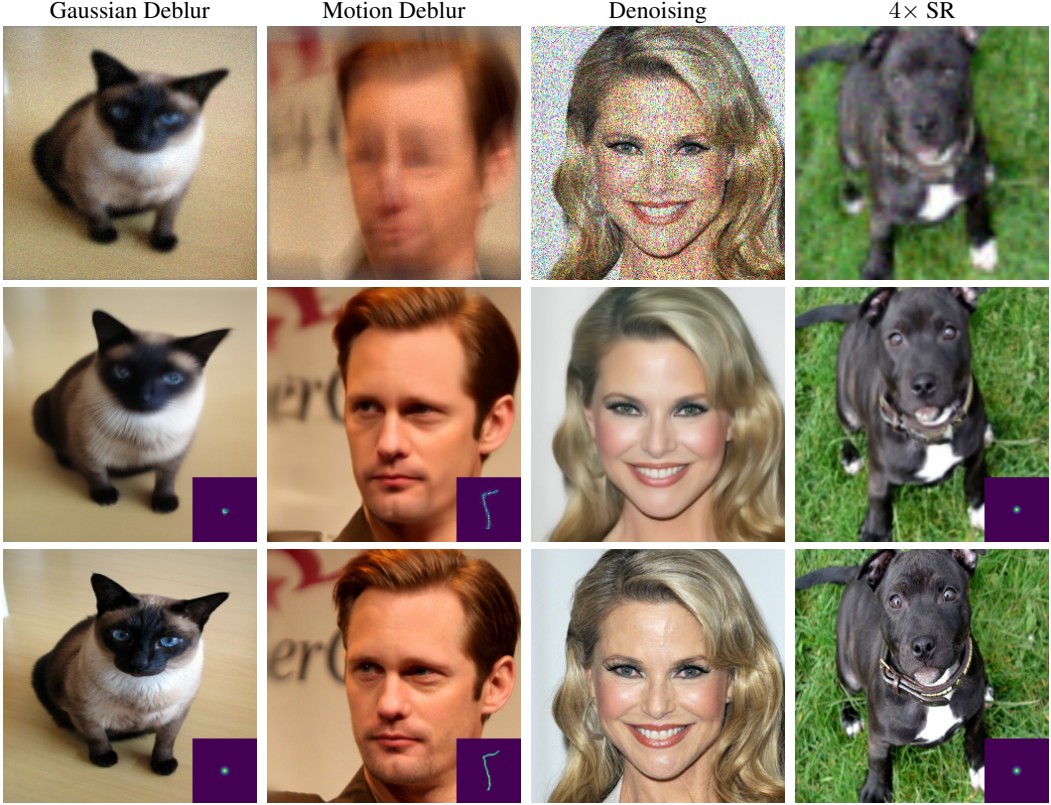

Figure 2: We demonstrate BIRD on several **blind** image restoration problems (*i.e.*, when the values of the degradation model are unknown): Gaussian deblurring, motion deblurring, superresolution (SR) (it includes additional Gaussian blur) and denoising with an unknown noise distribution. BIRD is applicable to a single degraded image and does not require re-training or fine-tuning of the prior model (we use a diffusion model). Although some of the generated degraded images use Gaussian blur, BIRD recovers a generic blur kernel (without making any Gaussianity assumption).

IR tasks based on the range-null space decomposition. The method works by refining only the null-space contents during the reverse diffusion process, to satisfy both data consistency and realness. Some recent examples of blind methods are GDP [6] and BlindDPS [1]. GDP leverages DDPM and solve inverse problems via hierarchical guidance and a patch-based method. BlindDPS proposes an extension of [2] to the case of blind deblurring. They train a score-based diffusion model for the blur operator. At test-time, they jointly optimize for both the sharp image and blur operator by running the reverse diffusion process. However, training a model for each new degradation operator can be time-consuming. Fast Diffusion EM [9] proposes a method for blind image deblurring. It applies the Expectation–Maximization (EM) algorithm after each reverse diffusion step to jointly update the image latent and blur kernel. Gibbsddrm [13] extends DDRM [8] to the blind case by adopting a Gibbs sampler to enable efficient sampling from the posterior distribution. In contrast to these works, in BIRD we introduce a blind image restoration method based on a novel diffusion inversion for DDIM. Our method is computationally efficient, as it requires only a few reverse diffusion steps, and generates realistic samples by only estimating the initial noise.

**Methods based on GAN inversion.** Since our proposed method is based on inverting a diffusion process, we also briefly review related work in the literature and discuss how it differs from our approach. Because Generative Adversarial Networks (GANs) were among the best image generation models, a number of methods [14, 15, 22, 23] inverts pre-trained GANs to solve image restoration problems. DGPGAN [14] performs the GAN inversion on a single new corrupted image. The authors shows that the inversion is not easily obtainable just with the direct optimization of the latent input to the frozen GANs model. Thus, they propose to also fine-tune the (unfrozen) GAN, while optimizing for the latent vector. [23] instead train an encoder to invert a pre-trained GANs on a

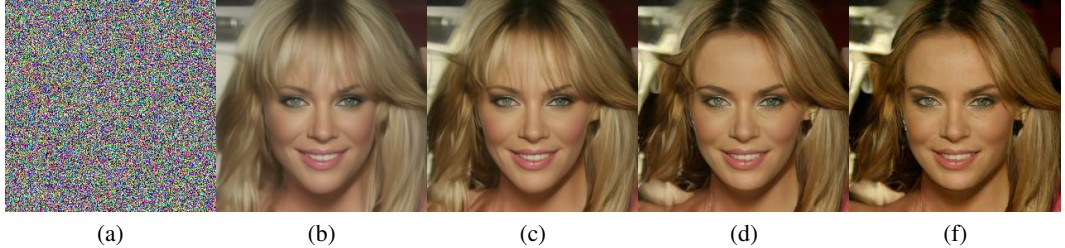

| (a) | (b) | (c) | (d) | (f) |

Figure 3: Illustration of our proposed accelerated image sampling of the pre-trained DDIM. (a) intial noise $x_T \sim \mathcal{N}(0, \mathbf{I})$ ($T = 1000$). (b), (c), (d), (e) and (f) are samples $x_0$ generated using DDIMReverse($x_T, \delta t$) with $\delta t = 100, 50, 20, 1$ respectively. Notice how the generated images are all realistic regardless of the choice of the step size $\delta t$.

dataset of corrupted images (through masking). At test time they directly apply the trained encoder and the GANs on a new corrupted image without further training. BIRD shares the aim of inverting a generative model with the above prior work. However, as demonstrated in the literature for inverting GANs models, the inversion of generative models is far from a straightforward task. Moreover, the inversion of GANs and diffusion models are fundamentally different in nature. For example, while GANs generate samples in "one forward pass" diffusion models require multiple (denoising) steps. Also, all IR methods based on GAN inversion either optimize the intermediate layers of the generator [4], train an auxiliary network (*e.g.*, an encoder), or fine-tune the pre-trained GAN network. In our method, the diffusion model is not fine-tuned/trained.

## 3 Image Restoration via BIRD

In this section, we introduce the classic Bayesian formulation of inverse problems applied to a single degraded image. An important component in this formulation is the characterization of the image prior, which we do via DDIM diffusion models.

### 3.1 Problem Formulation

Image restoration (IR) can be cast as a Maximum a Posteriori (MAP) optimization problem [24]

$$\hat{x} = \arg \min_x \log p(y|x) + \log p(x), \tag{1}$$

where $\hat{x} \in \mathbb{R}^{N_x \times M_x}$ is the restored image of size $N_x \times M_x$ pixels, $y \in \mathbb{R}^{N_y \times M_y}$ is the degraded image of size $N_y \times M_y$ pixels, where $p(y|x)$ is the so-called *likelihood* and $p(x)$ the *prior* distribution. Inverse problems describe IR tasks with a degradation operator $H_\eta : \mathbb{R}^{N_x \times M_x} \to \mathbb{R}^{N_y \times M_y}$ and a noise image $n \in \mathbb{R}^{N_y \times M_y}$, such that the observed degraded image $y$ can be written as

$$y = H_\eta(x) + n. \tag{2}$$

$\eta$ is a vector with all the parameters that define the operator $H_\eta$. IR tasks such as image denoising, deblurring and super-resolution can all be described using specific choices of $H_\eta$. For instance, in the case of image deblurring, the $H_\eta$ operator can be described as a convolution with a blur kernel $k$ such that $H_\eta(x) \doteq k * x$, where $*$ denotes the convolution operation. In the blind deblurring problem, one assumes that $H_\eta$ takes the form of a convolution, but without knowing its parameters $\eta$, which, in this case, are the values of the true blur kernel $k$. Thus, we also need to estimate the parameters $\eta$ as part of the optimization procedure. Only in the special case of image denoising we do use $H_\eta$ as the identity function. By assuming that the noise $n$ is zero mean Gaussian in the MAP formulation, one can rewrite eq. (1) as

$$\hat{x}, \hat{\eta} = \arg \min_{x \in \mathbb{R}^{N_x \times M_x}, \eta} \|y - H_\eta(x)\|^2 + \lambda \mathcal{R}(x), \tag{3}$$

where $\mathcal{R}(x)$ is also called a *regularization* term, and $\lambda > 0$ is a coefficient that regulates the interplay between the likelihood and the prior. We define $\Omega_x \in \mathbb{R}^{N_x \times M_x}$ as the (compact) set of degradation-free (realistic) images and we propose employing a formulation that implicitly assumes a uniform

---
**Algorithm 1** BIRD: Image Restoration
---
**Require:** Degraded image $y$, step size $\delta t$, the dimension of $x$ $d = N_x M_x$, learning rate $\alpha$, the stop threshold $\varepsilon$
**Ensure:** Return $\hat{x}_0$
1: Initialize $x_T^0 \sim \mathcal{N}(0, \mathbf{I})$ and $H_{\eta^0}$ with random parameters
2: **while** $k : 1 \to N$ and $\mathcal{L}_{IR}(x_0^k, H_{\eta^k}) > \varepsilon$ **do**
3:     $x_0^k = \text{DDIMReverse}(x_T^k, \delta t)$
4:     $x_T^{k+1} = x_T^k - \alpha \nabla_{x_T} \mathcal{L}_{IR}(x_0^k, H_{\eta^k})$
5:     $x_T^{k+1} = \frac{x_T^{k+1}}{\|x_T^{k+1}\|} \sqrt{d}$
6:     $\eta^{k+1} = \eta^k - \alpha \nabla_\eta \mathcal{L}_{IR}(x_0^k, H_{\eta^k})$
7: **end while**
8: **return** $\hat{x}_0 = \text{DDIMReverse}(x_T^N, \delta t)$
---

---
**Algorithm 2** DDIMReverse ( $x_T$ , $\delta t$)
---
**Require:** : Initial noise $x_T \sim \mathcal{N}(0, \mathbf{I})$, step size $\delta t$
1: $t = T$
2: **while** $t > 0$ **do**
3:     $\hat{x}_{0|t} = (x_t - \sqrt{1 - \bar{\alpha}_t}\epsilon_\theta(x_t, t))/\sqrt{\bar{\alpha}_t}$
4:     $x_{t-\delta t} = \sqrt{\bar{\alpha}_{t-\delta t}}\hat{x}_{0|t} + \sqrt{1 - \bar{\alpha}_{t-\delta t}} \cdot \frac{x_t - \sqrt{\bar{\alpha}_t}\hat{x}_{0|t}}{\sqrt{1-\bar{\alpha}_t}}$
5:     $t \leftarrow t - \delta t$
6: **end while**
7: return $\hat{x}_0$
---

prior $p(x) \doteq p_U(x)$ on $\Omega_x$. That is, $\mathcal{R}(x) = -\log(p_U(x)) = -\log(c \cdot \mathbb{1}_{\Omega_x}(x))$, where $c$ is a normalizing constant and $\mathbb{1}_{\Omega_x}(x)$ is 1 if $x$ is in the support of $\Omega_x$ and 0 otherwise.

This results in the following simplified formulation

$$\hat{x}, \hat{\eta} = \underset{x \in \Omega_x, \eta}{\arg\min} \|y - H_\eta(x)\|^2. \tag{4}$$

To ensure that $x \in \Omega_x$, we parameterize $x$ via the initial noise $z$ of a pre-trained diffusion model $g : \mathbb{R}^{N_x \times M_x} \to \mathbb{R}^{N_x \times M_x}$ trained by mapping noise samples from the high-density region of the standard Normal distribution (that we denote $\Omega_z$) to the domain of degradation-free (realistic) images $\Omega_x$. In other words, we assume that $\Omega_x = \{g(z)\}|_{z \in \Omega_z}$. The optimization objective eq. (4) becomes then

$$\hat{z}, \hat{\eta} = \underset{z \in \Omega_z, \eta}{\arg\min} \|y - H_\eta(g(z))\|^2, \tag{5}$$

with $\hat{x} = g(\hat{z})$. In high dimensions, $\frac{1}{N_x M_x}\|z\|^2 \approx \mathbb{E}[zz^\top] = \text{variance}(z) = 1$ as $N_x M_x \mapsto \infty$. In fact, most of the density of a high-dimensional Normal random variable is around $\|z\|^2 = N_x M_x$ [12, 20]. Therefore, we propose to approximate the original problem (3) with

$$\hat{z}, \hat{\eta} = \underset{z : \|z\|^2 = N_x M_x, \eta}{\arg\min} \|y - H_\eta(g(z))\|^2. \tag{6}$$

## 3.2 An Efficient Diffusion Inversion

In problem (6), we implement the function $g$ by adopting a pre-trained diffusion model. We choose the Denoising Diffusion Implicit Model (DDIM) [17] and set it as a fully deterministic diffusion process. To solve problem (6), we propose an optimization-based iterative procedure. We jointly estimate a realistic image $x$ (*i.e.*, with a high $p(x)$) and the forward degradation model $H_\eta$ such that $H_\eta(x) \approx y$. In order to find $x$, we explicitly exploit the correspondence between image samples $x$ and their associated initial latent image $z$ in DDIMs, in contrast to the mappings used in prior work [8, 11, 21]. Thus, we aim to find the initial noise sample $z$ that can generate the image $x$ when applied to DDIM. To keep the notation consistent with the DDIM formalism, we denote images $x$ with $x_0$ and the samples $z$ with $x_T$, where $T$ is the number of iterations in the diffusion model. To make our presentation self-contained, we briefly revise the notation and notions of diffusion models.

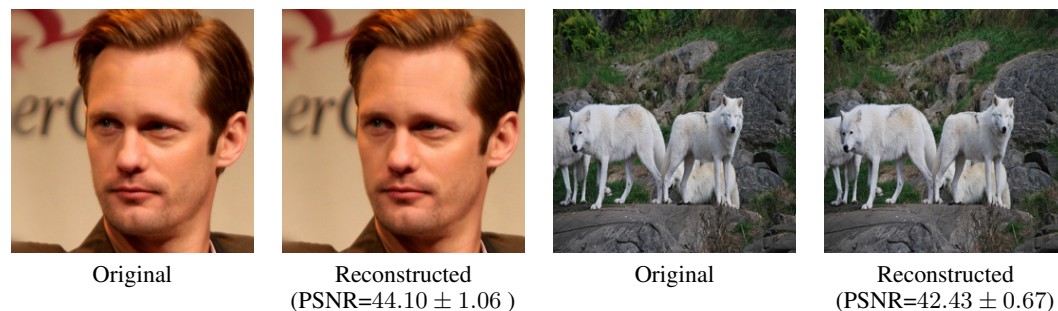

| Original | Reconstructed (PSNR=$44.10 \pm 1.06$ ) | Original | Reconstructed (PSNR=$42.43 \pm 0.67$) |

Figure 4: Reconstruction results using BIRD on samples from CelebA and ImageNet validation datasets. The PSNR mean and standard deviation are computed over 10 runs.

### 3.2.1 Background: Denoising Diffusion Probabilistic and Implicit Models

Denoising Diffusion Probabilistic Models (DDPM) [7] leverage diffusion processes in order to generate high quality image samples. The aim is to reverse the forward diffusion process that maps images to noise, either by relying on a stochastic iterative denoising process or by learning the explicit dynamics of the reverse process, *e.g.*, through an ODE [17]. More precisely the forward diffusion process maps an image $x_0 \sim p(x)$ to a zero-mean Gaussian $x_T \sim \mathcal{N}(0, \mathbf{I})$ by generating intermediate images $x_t$ for $t \in [1, T]$ which are progressively noisier versions of $x_0$. DDPM [7] adopts a Markovian diffusion process, where $x_t$ only depends on $x_{t-1}$. Given a non-increasing sequence $\alpha_{1:T} \in (0, 1]$, the joint and marginal distributions of the forward diffusion process are described by

$$q(x_{1:T}|x_0) = \prod_{t=1}^{T} q(x_t|x_{t-1}), \text{ where } q(x_t|x_{t-1}) = \mathcal{N}\left(\sqrt{\alpha_t}x_{t-1}, \left(1 - \alpha_t\right)\mathbf{I}\right), \tag{7}$$

which implies that we can sample $x_t$ simply by conditioning on $x_0$ with $\bar{\alpha}_t = \prod_{s \leq t} \alpha_s$ via

$$q(x_t|x_0) = \mathcal{N}\left(\sqrt{\bar{\alpha}_t}x_0, (1 - \bar{\alpha}_t)\mathbf{I}\right). \tag{8}$$

To invert the forward process, one can train a model $\epsilon_\theta$, with parameters $\theta$, to minimize the objective

$$\min_{\theta} \quad \mathbb{E}_{t \sim \mathcal{U}(0,1); x_0 \sim q(x); \epsilon \sim \mathcal{N}(0,\mathbf{I})} \left[ \|\epsilon - \epsilon_\theta(\sqrt{\bar{\alpha}_t}x_0 + \sqrt{1 - \bar{\alpha}_t}\epsilon, t)\|^2 \right]. \tag{9}$$

Given the initial noise $x_T$, image samples $x_0$ are obtained by iterating for $t \in [1, T]$ the denoising update

$$x_{t-1} = \frac{1}{\sqrt{\alpha_t}} \left( x_t - \epsilon_\theta(x_t, t) \times \frac{(1 - \alpha_t)}{\sqrt{1 - \bar{\alpha}_t}} \right) + \sigma_t z, \tag{10}$$

with $z \sim \mathcal{N}(0, \mathbf{I})$, and $\sigma_t^2 = \frac{1 - \bar{\alpha}_{t-1}}{1 - \bar{\alpha}_t}(1 - \alpha_t)$.

The authors of [17] point out that the quality of generated images improves as the total number of denoising steps $T$ increases. Thus, the inference loop using eq. (10) becomes computationally expensive. To reduce the computational cost, they propose to use instead

$$q(x_{1:T}|x_0) \; = \; q(x_T|x_0) \prod_{t=2}^{T} q(x_{t-1}|x_t, x_0), \tag{11}$$

a Denoising Diffusion Implicit Model (DDIM) [17], which foregoes the Markovian assumption in favor of a diffusion process where $q(x_T|x_0) = \mathcal{N}\left(\sqrt{\bar{\alpha}_T}x_0, (1 - \bar{\alpha}_T)\mathbf{I}\right)$ and

$$q(x_{t-1}|x_t, x_0) = \mathcal{N}\left(\sqrt{\bar{\alpha}_{t-1}}x_0 + \sqrt{1 - \bar{\alpha}_{t-1} - \sigma_t^2} . \frac{(x_t - \sqrt{\bar{\alpha}_t}x_0)}{\sqrt{1 - \bar{\alpha}_t}}, \sigma_t^2\mathbf{I}\right). \tag{12}$$

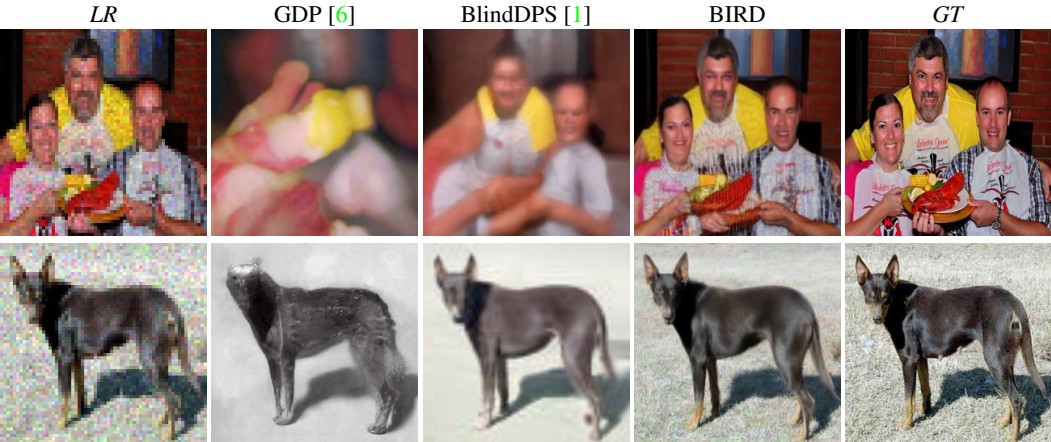

Figure 5: Qualitative comparisons of $4\times$ SR on ImageNet. Each column shows two examples. From left to right: input low-resolution images, GDP [6], BlindDPS [1], BIRD (our method), and the ground truth (GT) high-resolution and noise-free image.

When $\sigma_t = 0$ for all $t$, the diffusion process is **fully deterministic**. This means that when we start from the same noise sample $x_T$ we obtain the same generated image sample. Given $x_t$, one can first predict the denoised observation $\hat{x}_0$, which is a prediction of $x_0$ given $x_t$

$$\hat{x}_0 = \frac{(x_t - \sqrt{1 - \bar{\alpha}_t}\epsilon_\theta(x_t, t))}{\sqrt{\bar{\alpha}_t}}. \tag{13}$$

Then, we can predict $x_{t-1}$ from $x_t$ and $\hat{x}_0$ using eq. (12) by setting $\sigma_t = 0$

$$x_{t-1} = \sqrt{\bar{\alpha}_{t-1}}\hat{x}_0 + \sqrt{1 - \bar{\alpha}_{t-1}}\hat{\omega} \tag{14}$$

with $\hat{\omega} = \frac{x_t - \sqrt{\bar{\alpha}_t}\hat{x}_0}{\sqrt{1-\bar{\alpha}_t}}$ a direction pointing to $x_t$. [17] shows that this formulation allows DDIM to use fewer time steps at inference by directly predicting $x_{t-\tau}$ with $\tau > 1$, which results in a more computationally efficient process using

$$x_{t-\tau} = \sqrt{\bar{\alpha}_{t-\tau}}\hat{x}_0 + \sqrt{1 - \bar{\alpha}_{t-\tau}}\hat{\omega}. \tag{15}$$

### 3.2.2 Accelerated DDIM Sampling

As previously mentioned, we adopt DDIMs as our pre-trained generative process. Here, we explore the advantage of DDIMs of allowing fewer steps than in other diffusion models during the sampling process, as discussed in section 3.2.1. We start from $x_T \sim \mathcal{N}(0, \mathbf{I})$. Instead of denoising $x_T$ iteratively for all the steps used in the pre-training, we make larger steps by using intermediate estimates of $\hat{x}_{0|t}$ from $x_t$ using the pre-trained DDIM model $\epsilon_\theta$ via

$$\hat{x}_{0|t} = \frac{x_t - \sqrt{1 - \bar{\alpha}_t}\epsilon_\theta(x_t, t)}{\sqrt{\bar{\alpha}_t}}. \tag{16}$$

We define the hyper-parameter $\delta t$ that controls the number of denoising steps, and we can directly jump to estimate $x_{t-\delta t}$ from $\hat{x}_{0|t}$ and $x_t$ using

$$x_{t-\delta t} = \sqrt{\bar{\alpha}_{t-\delta t}}\hat{x}_{0|t} + \sqrt{1 - \bar{\alpha}_{t-\delta t}} \cdot \frac{x_t - \sqrt{\bar{\alpha}_t}\hat{x}_{0|t}}{\sqrt{1 - \bar{\alpha}_t}}. \tag{17}$$

A larger step size $\delta t$ allows us to favor speed, while a lower one favors precision or fidelity. This iterative procedure, which we summarize in Algorithm 2, results in an estimate of $x_0$ that is differentiable in $x_T$. We denote by DDIMReverse$(., \delta t)$ the mapping function between $x_T$ and $x_0$ that is parameterized with the step size $\delta t$ (this is essentially our choice of $g$ in problem (6)). Figure 3 shows sampled images $x_0$ using different $\delta t$ starting from the same initial noise $x_T$ through DDIMReverse$(., \delta t)$.

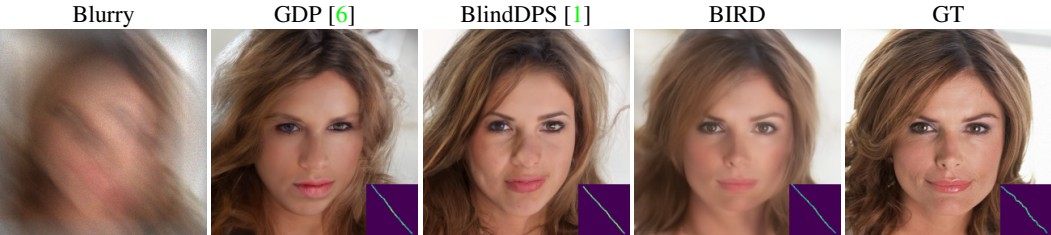

| Blurry | GDP [6] | BlindDPS [1] | BIRD | GT |

Figure 6: Qualitative comparisons of Gaussian deblurring on CelebA. From left to right: input blurry image, GDP [6], BlindDPS [1], BIRD (our method), and the ground truth sharp image.

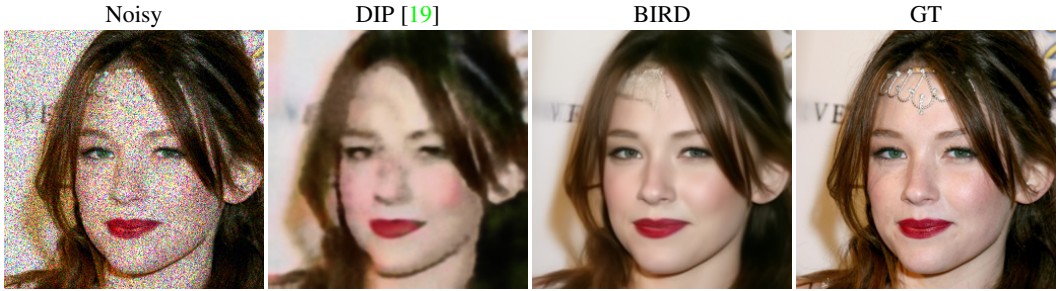

| Noisy | DIP [19] | BIRD | GT |

Figure 7: Qualitative comparisons of image denoising on CelebA. From left to right: input noisy image, GDP [6], BlindDPS [1], BIRD and the original image (GT).

### 3.2.3 BIRD: Blind Image Restoration via Fast Diffusion Inversion

BIRD performs an iterative minimization of problem (6) with a running index $k$, which we append as a superscript to the estimated unknowns. It starts from an initial noise instance $x_T^0 \sim \mathcal{N}(0, \mathbf{I})$ and a parametric degradation model $H_{\eta^0}$, where $\eta^0$ is randomly initialized. At each iteration $k$, we first compute our estimate of the clean image $x_0^k$ by mapping $x_T^k$ through DDIMReverse$(., \delta t)$

$$x_0^k = \text{DDIMReverse}(x_T^k, \delta t). \tag{18}$$

We then compute the restoration loss

$$\mathcal{L}_{\text{IR}}(x_0^k, H_\eta) = \|y - H_\eta(x_0^k)\|^2. \tag{19}$$

Given an initial noise $x_T^k$ and $H_{\eta^k}$, the optimization iteration is simply derived through gradient descent with a learning rate $\alpha$

$$x_T^{k+1} = x_T^k - \alpha \nabla_{x_T} \mathcal{L}_{\text{IR}}(x_0^k, H_{\eta^k}) \quad \text{and} \quad \eta^{k+1} = \eta^k - \alpha \nabla_\eta \mathcal{L}_{\text{IR}}(x_0^k, H_{\eta^k}). \tag{20}$$

To impose the normalization constraint on $x_T$ we set its Euclidean norm to $\|x_T\| = \sqrt{N_x M_x}$ via

$$x_T^{k+1} = \frac{x_T^{k+1}}{\|x_T^{k+1}\|} \sqrt{N_x M_x}. \tag{21}$$

After $N$ iterations, or when $\mathcal{L}_{\text{IR}}(x_0^N, H_{\eta^N})$ becomes small enough, we deem that BIRD has converged. The restored image $\hat{x}_0$ is then generated by $\hat{x}_0 = \text{DDIMReverse}(x_T^N, \delta t)$, which always ensures that $\hat{x}_0$ is within the manifold of realistic images, as already shown in Figure 3. Moreover, in Figure 4 we show two examples of the reconstruction with $H_\eta(x) = x$ and with $y$ un-corrupted noise-free images from the validation dataset of ImageNet [5] and CelebA [10]. We summarize this iterative procedure in Algorithm 1.

## 4 Experiments

### 4.1 Image Restoration Tasks

We showcase BIRD on four different image restoration tasks: Gaussian and motion deblurring, denoising, and image super-resolution (SR). For Gaussian deblurring, we use an anisotropic Gaussian

Table 1: Quantitative evaluation of several inverse problems on the CelebA validation dataset. The best and second best methods are indicated in bold and underlined respectively.

| Method | Motion Deblur | | Gaussian Deblur | | 8× SR | | Denoising | |
|---|---|---|---|---|---|---|---|---|
| | PSNR ↑ | LPIPS ↓ | PSNR ↑ | LPIPS ↓ | PSNR ↑ | LPIPS ↓ | PSNR ↑ | LPIPS ↓ |
| BlindDPS [1] | 23.15 | 0.281 | 23.56 | 0.257 | 21.82 | 0.345 | - | - |
| DIP [19] | - | - | - | - | 18.64 | 0.415 | 24.57 | 0.282 |
| Fast Diffusion EM [9] | 23.18 | 0.284 | 24.52 | 0.235 | - | - | - | - |
| GibbsDDRM [13] | 22.94 | 0.314 | 23.57 | 0.266 | - | - | - | - |
| DGPGAN [14] | 22.23 | 0.304 | 20.65 | 0.378 | 19.83 | 0.372 | - | - |
| GDP [6] | 22.49 | 0.314 | 22.53 | 0.304 | 20.78 | 0.357 | - | - |
| BIRD | **23.76** | **0.263** | **24.67** | **0.225** | **22.75** | **0.306** | **28.46** | **0.227** |

Table 2: Quantitative evaluation of several inverse problems on the ImageNet validation dataset. The best and second best methods are indicated in bold and underlined respectively.

| Method | Gaussian Deblur | | 4× SR | |
|---|---|---|---|---|
| | PSNR ↑ | LPIPS ↓ | PSNR ↑ | LPIPS ↓ |
| BlindDPS [1] | **24.28** | **0.296** | 21.36 | 0.374 |
| DIP [19] | - | - | 18.67 | 0.456 |
| DGPGAN [14] | 21.67 | 0.384 | 19.58 | 0.461 |
| GDP [6] | 22.45 | 0.368 | 20.24 | 0.435 |
| BIRD | 23.76 | 0.321 | **22.15** | **0.354** |

kernel. We follow the procedure described in [16] and use the same feed-forward network for the kernel estimation. For all tasks, we note that we consider the noisy case with a noise standard deviation of $\sigma \approx 0.05$. For image denoising, we introduce a combination of Gaussian (signal-independent) and speckle (signal-dependent) noises simulating common sources of noise in cameras, namely shot noise and dead (or hot) pixels respectively. For all our experiments, we use $\delta t = 100$ and a maximum number of iterations $N = 200$. Adam is adopted as an optimizer with a 0.003 learning rate. We evaluate our method both on the validation datasets of ImageNet [5] and CelebA [10] at $256 \times 256$ pixel resolution. We follow [2, 3] and report the obtained LPIPS [25] and PSNR for each experiment.

## 4.2 Qualitative Results

Figures 5, 6, 7 show non cherry-picked examples of ImageNet and CelebA images restored in the cases of $4\times$ super-resolution, Gaussian deblurring and denoising. We also compare our work with GDP [6] and BlindDPS [1]. BIRD seems to recover always more realistic, consistent and accurate images than the other competing methods.

## 4.3 Quantitative Results

Tables 1 and 2 report the performance of BIRD as well as DIP [19], DGPGAN [14], DPS [2], GDP [6] for CelebA and ImageNet, respectively. For both datasets, we observe that our method achieves state of the art results both in terms of PSNR and LPIPS metrics.

## 4.4 Ablations

We also compare the runtime performance of BIRD to other state-of-the-art methods and evaluate the impact of the step size $\delta t$. All experiments are carried out on a GeForce GTX 1080 Ti.
**Effect of the step size $\delta t$:** The step size $\delta t$ defines a trade-off between speed and accuracy. Table 3 shows the PSNR, LPIPS and runtime for image denoising on CelebA. We observe that the reconstruction accuracy of BIRD is not too sensitive to this parameter, while the runtime is linearly affected. $\delta t = 100$ gives a good trade-off between the speed and the reconstruction accuracy.
**Memory usage and Computational Cost:** In Table 4, we report the memory usage and the runtime per image of different blind deblurring methods. We also show as a reference a *Naive Inversion*, which performs the inversion by unrolling all the steps of the pre-trained diffusion model. This case

Table 3: Effect of the step size $\delta t$ on the visual quality and the runtime of denoising on CelebA.

| Step size ($\delta t$) | PSNR ↑ | LPIPS ↓ | Time[s] ↓ |
|---|---|---|---|
| 50 | 28.74 | 0.218 | 412 |
| 100 | 28.67 | 0.224 | 234 |
| 200 | 28.45 | 0.237 | 110 |

Table 4: Runtime (in seconds per processed image) comparison of blind deblurring methods on CelebA.

| Method | Time[s] | Memory[GB] |
|---|---|---|
| DGPGAN [14] | 190 | 0.9 |
| GDP [6] | 118 | 1.1 |
| *Naive Inversion* | 5700 | 2.2 |
| BlindDPS [1] | 270 | 6.1 |
| BIRD | 234 | 1.2 |

shows how BIRD results in a significantly faster execution. Despite being an iterative method, BIRD has a reasonable runtime making it a practical IR method. The reported speed uses $\delta t = 100$.

### 4.5 Limitations and Broader Impacts

We have evaluated our method on degradation operators with a given explicit parametrized form (albeit with unknown parameters). It would be interesting to further investigate the use of BIRD on problems where an explicit form of the degradation operator is difficult to have, such as in image deraining and dehazing. In terms of broader impacts, BIRD leverages pre-trained denoising diffusion models with both their advantages and pitfalls. It takes advantage of their strong prior, but may also be affected by their data-induced biases.

## 5 Conclusion

We have introduced BIRD, a novel robust, accurate and fast framework for solving general blind image restoration tasks by using pre-trained diffusion generative models as learned priors. Our method exploits the deterministic correspondence between noise and images in DDIM by casting the inverse restoration problem as a latent estimation problem. Our framework does not require training networks on specific task, but can instead be directly applied to new images and new degradation models. We leverage the capability of DDIM to skip ahead in the forward diffusion process and provide an efficient diffusion inversion in the context of image restoration. We demonstrate that BIRD achieves state of the art performance on blind image restoration tasks including Gaussian deblurring, motion deblurring, super-resolution and denoising.

**Acknoweledgements.** We acknowledge the support of the SNF project number 200020_200304.

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

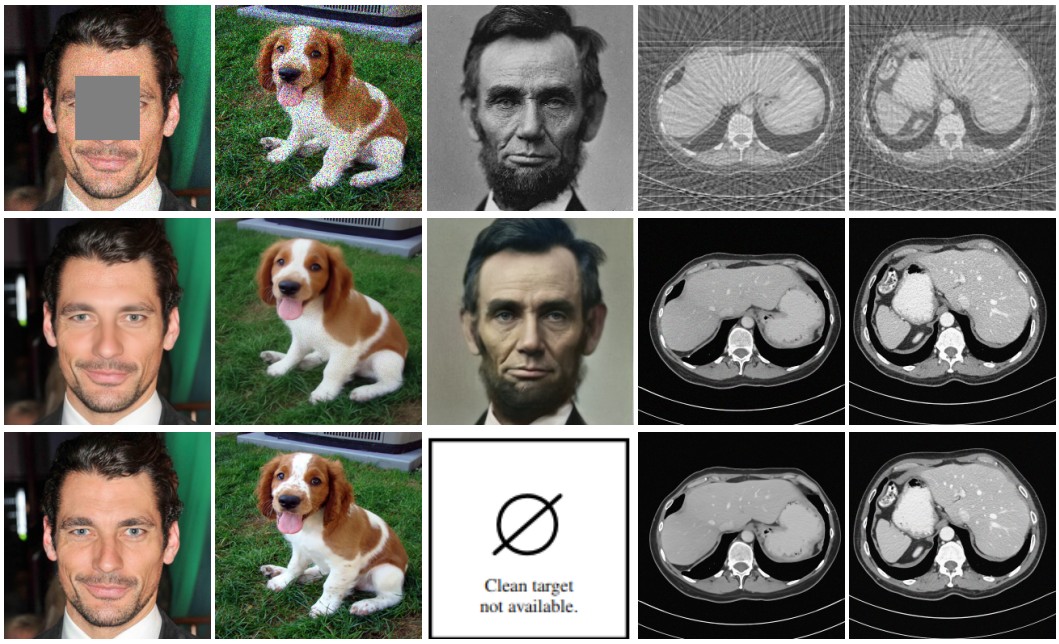

Figure 8: BIRD can also be applied to non-blind tasks. In particular, notice the special case of sparse-view CT image reconstruction (rightmost two columns), where the measurements are not in the image domain (on the top row we show the reprojections of the measurements). First row: Measurements. Second row: BIRD reconstructions. Third row: Ground-truth data.

# A    Appendix / supplemental material

## A.1    Application of BIRD to non-blind cases

Naturally, BIRD can also be applied to non-bind cases, *i.e.*, when the degradation operator is completely known. In Figure 8, we show the application of BIRD to some non-blind image restoration tasks such as image inpainting and sparse-view CT image reconstruction.

## A.2    Robustness of BIRD to severe degradation

BIRD is also robust to severe degradation. In Figure 9, we show that BIRD is more robust in the case of face hallucination ($\times 16$ super-resolution). BIRD outperforms the other methods in terms of visual quality as well as faithfulness.

## A.3    Robustness of BIRD to errors in the degradation operator estimate

Non-blind methods generally rely on off-the-shelf methods to estimate the degradation operator. These off-the-shelf methods are not 100% accurate. We simulate a small error in the degradation operator and we use the simulated kernel (plus this error) to generate the restored image for the non-blind DPS [2] and BIRD. As shown in Figure 10, BIRD produces better quality outputs compared to the non-blind method DPS when using the kernel with a slightly simulated error.

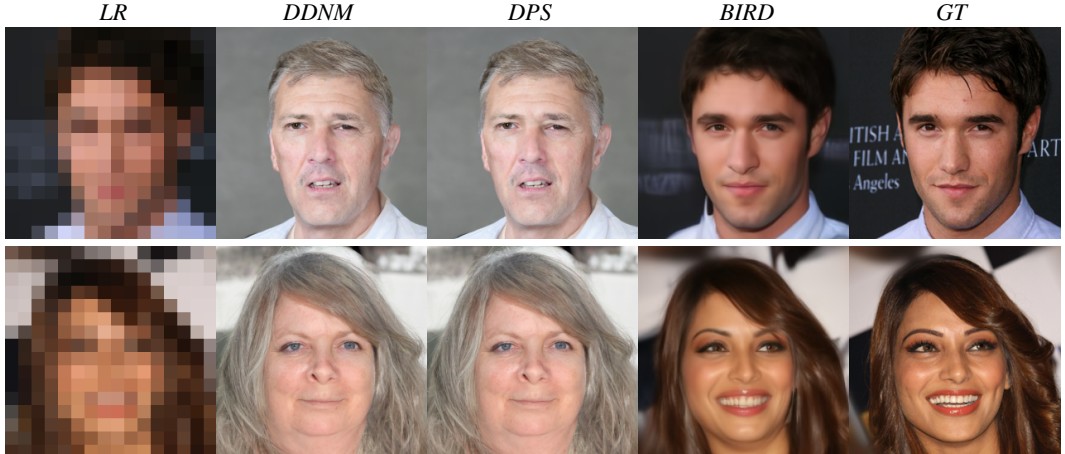

Figure 9: Robustness to severe degradation (Face hallucination, ×16 superresolution). BIRD is more robust to severe degradations than competing methods, both in terms of visual image quality and faithfulness.

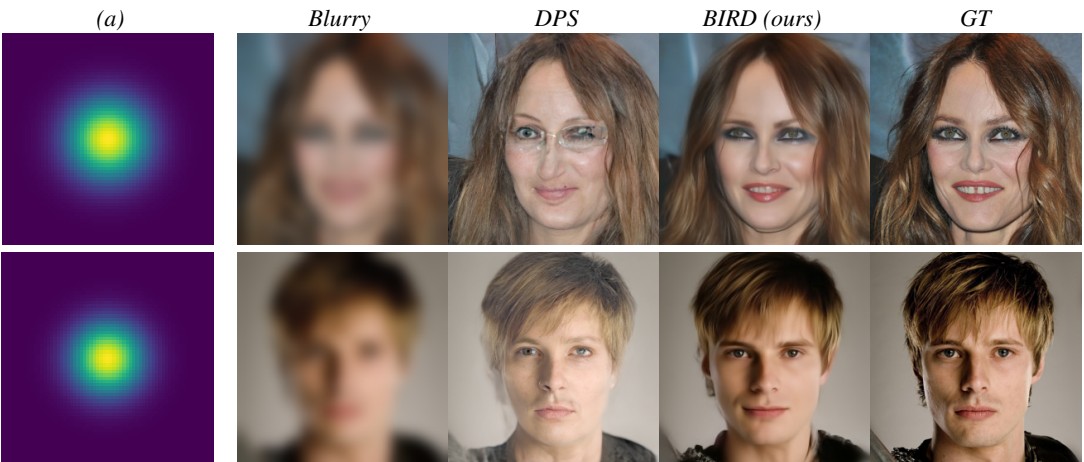

Figure 10: Robustness to errors in the degradation operator estimate. First column: ground-truth kernel (std=8.5) and a kernel with a slightly simulated error (std=8). Second column: blurry images using the kernel with a slightly simulated error. Third and fourth columns: Output of DPS and BIRD. Fifth column: Ground-truth.

