# OpenReview forum: "Blind Image Restoration via Fast Diffusion Inversion"
_NeurIPS.cc/2024/Conference — NeurIPS 2024 poster_

### Official Review · Reviewer_cmXb · 2024-07-10

**Soundness:** 3
**Presentation:** 3
**Contribution:** 2
**Rating:** 4
**Confidence:** 5

**Summary:**

The paper introduces a blind image restoration method based on DDIM, which iteratively optimizes the initial noises and the degradation model parameters through the restoration loss of reconstructing the degraded image. As a result, the restored image remains on the data manifold of the pretrained diffusion model. The experiments on three image restoration tasks demonstrate the feasibility of the proposed method.

**Strengths:**

1. The method do not alter the reverse sampling, and could generate images that lie on the data manifold of diffusion model at every iteration.
2. The method is adaptable to multiple image restoration tasks, such as deblurring, super-resolution, and JPEG de-artifacting, without requiring model retraining or fine-tuning.

**Weaknesses:**

1. The inversion of generative models has been extensively used in both GAN and diffusion models. This work does not sufficiently distinguish itself from existing methods.
2. The optimizable degradation model has been presented in the paper of GDP[1], and the iterative optimization of the degradation model and the restored image also has been adopted in the paper of TAO[2]. The authors have not adequately addressed how their method differs from these existing approaches.
3. The paper claims that the method is computationally efficient, yet the timing results presented in Table 2 do not support this claim. In addition, despite the authors' assertion that their method achieves state-of-the-art results, the performance metrics in Table 4 do not substantiate this claim.
[1] Fei B, Lyu Z, Pan L, et al. Generative diffusion prior for unified image restoration and enhancement[C]//Proceedings of the IEEE/CVF Conference on Computer Vision and Pattern Recognition. 2023: 9935-9946.
[2] Gou Y, Zhao H, Li B, et al. Test-Time Degradation Adaptation for Open-Set Image Restoration[C]//Forty-first International Conference on Machine Learning.

**Questions:**

Please address the concerns in the above Weaknesses.

**Limitations:**

Please see the above Weaknesses.

---

> ### Author Rebuttal · Authors · 2024-08-05
>
> Thank you for your feedback. Please see the detailed response below.
>
>
> ### Q1. Further distinction from existing methods based on inversion
> First, we point that there is a a fundamental difference between *inverting a clean image* (mostly useful for applications like image editing), where we aim for the same image, and *inverting a degraded image* (useful for IR tasks), where we aim for the degradation-free image. Indeed, a lot of the literature related to the inversion of generative models applies to the clean case, which is different from what IR tasks aim to solve. Here, we further emphasize the key differences between BIRD and other IR methods based on diffusion and GAN  inversion.
>
>  *Diffusion-based approaches*
>
> We can fairly claim that BIRD is the *first* diffusion-based approach to invert a *degraded* image. By inversion, we mean obtaining the initial Gaussian noise $x_T$, which, when applied to the diffusion model, yields the clean image. There are a few methods that apply diffusion inversion to a clean image in the context of image editing, such as [3, 4]. Based on the assumption that the ODE process can be reversed in the limit of small steps, these methods run the diffusion in the reverse direction from $z_0$ to \$z_T $ (from image space to Gaussian noise), rather than from $z_T$ to $ z_0$ to obtain the inverted latent. Unlike these methods, which solve a different task, BIRD is fundamentally different as it employs a gradient descent optimization approach, whereas [3, 4] do not. Moreover, applying those methods to a degraded image results in the same degraded image.
>
>
> *GAN-based approaches*
>
> The IR methods based on GAN inversion can be mainly classified into two approaches: 1) learning-based methods such as [5, 6], which train an encoder to invert the GAN. These methods are clearly distinguishable from BIRD, as BIRD is not a learning-based method and does not involve training. 2) Optimization-based methods such as [7, 8]. Although both these methods and BIRD are gradient descent optimization methods, they differ in key ways: 1) They fine-tune the GAN or train an auxiliary network, whereas BIRD involves no fine-tuning or training. 2) A diffusion model differs from a GAN in aspects such as iterative sampling versus a single forward pass and inference speed. Simply applying a GAN-style inversion to a diffusion model would take on the order of hours, which is impractical.
>
> Although many methods apply GAN inversion to IR tasks, the literature still lacks a diffusion inversion method despite its potential advantages. We believe that BIRD fills this gap and brings all the benefits of diffusion combined with inversion. In our humble opinion, BIRD can open new perspectives on the applicability of diffusion models in IR tasks and encourage further exploration of diffusion inversion for IR applications.
>
>
> ### Q2. Difference with GDP[1] and TAO[2], the iterative optimization of the degradation model and the restored image
>
> The idea of jointly optimizing the degradation model and the clean image (or doing it iteratively) is not new and is neither a contribution of ours nor of GDP [1]. The contributions of GDP and BIRD specifically lie in *how* to use the diffusion prior to obtain the clean image. In this regard, BIRD is fundamentally different from all other methods, particularly from GDP [1]. As stated in our contributions (lines 65-67), BIRD is the *first* to frame the IR problem as a latent (initial noise) optimization problem within the context of diffusion models. Based on this fundamental difference, there are many algorithmic differences between GDP [1] and BIRD. For example, our method is a pure *optimization-based* approach, where we optimize a well-defined variable (the initial noise $z_T$ ), while GDP is a *sampling* method. Moreover, in GDP, the degradation model is updated after each diffusion step while in BIRD, it is updated only after the hole diffusion process.
>
> BIRD is also significantly different from TAO [2] because TAO is an open-set method, whereas BIRD is a zero-shot method. Additionally, BIRD focuses on diffusion inversion, while TAO [2] does not.
>
>
> ###  Q3. Computational efficiency and state of the art?
>
> BIRD consistently yields the best quantitative results in Table 1 and Table 4, except in two cases. We believe that a state-of-the-art method does not necessarily outperform all other methods across all datasets and metrics. This is the case, for example, with GDP [2] and DPS [9]. Regarding computational efficiency, BIRD is slower than GDP but significantly better in terms of image quality. It is faster than the state-of-the-art BlindDPS while requiring five times less memory. We believe that BIRD presents the best trade-off between image quality and computational efficiency.
>
>
> [3] Prompt-to-Prompt Image Editing with Cross-Attention Control. arXiv 2022
>
> [4] Null-text Inversion for Editing Real Images using Guided Diffusion Models. CVPR 2024
>
> [5] High-fidelity image inpainting with gan inversion. ECCV 2022
>
> [6] Gan prior embedded network for blind face restoration in the wild. CVPR 2021
>
> [7] Robust unsupervised stylegan image restoration. CVPR 2023
>
> [8] Exploiting deep generative prior for versatile image restoration and manipulation. PAMI 2021
>
> [9] Diffusion Posterior Sampling for General Noisy Inverse Problems. ICLR 2023

---

> ### Comment · Reviewer_cmXb · 2024-08-14
> **Official Comment by Reviewer cmXb**
>
> I appreciate the authors' efforts in the rebuttal, and keep my original score since I reserve my opinion on the first two weaknesses.

---

> ### Author Response · Authors · 2024-08-14
>
> First, we thank the reviewer for their response.
>
> The reviewer reserves their opinion on the first two points without providing further justification.
>
> Regarding the first point (distinction with inversion-based IR methods), we stated that BIRD is the first IR method based on diffusion inversion. We would be grateful if the reviewer could point out any work that applies diffusion inversion for IR. Additionally, we mentioned that a GAN-style inversion approach will not work for diffusion, as in contrast to GANs, one clear difference is that diffusion is much more computationally demanding.
>
> Regarding the second point (differences with GDP [1] and TAO [2]), we noted that a key difference is that GDP [1] is a sampling method (it alters the reverse sampling of diffusion by adding a projection-based step to ensure consistency), whereas BIRD is an optimization-based method (it does not alter the reverse sampling). For TAO, a significant difference is that TAO is an open-set method, while BIRD is zero-shot. Figure 1 in TAO [2] illustrates this clearly.

---

### Official Review · Reviewer_ebAr · 2024-07-11

**Soundness:** 2
**Presentation:** 2
**Contribution:** 2
**Rating:** 4
**Confidence:** 4

**Summary:**

The authors propose a novel approach to solving image restoration problems using diffusion models, termed BIRD.
Unlike previous approaches, BIRD alternates between optimizing a parameterized forward operator and the initial latent variable of a DDIM to address various IR problems in a blind manner (i.e., with an unknown forward operator). This iterative process involves mapping the current latent variable to its outcome in the DDIM via the DDIMReverse algorithm and alternately taking a gradient step of the measurement error with respect to the current latent variable and the forward operator's parameters.

**Strengths:**

This paper presents an original approach by leveraging pretrained diffusion models to address blind inverse problems in image restoration. The simplicity and novelty of the method lie in its creative combination of well-known techniques, making it a noteworthy contribution to the field.

The quality of the theoretical analysis, particularly in sections 3.2 - 3.2.2, is commendable as it is grounded in related work, providing a solid foundation for the proposed method. The results presented demonstrate the method's potential and effectiveness in addressing the stated problems.

The clarity of the paper is generally good, with the authors providing a comprehensive introduction and detailed descriptions of the algorithms. The inclusion of specific equations and theoretical justifications helps in understanding the underlying principles of the method. The paper's structure and flow, from the introduction to the results, are logically organized, aiding in the reader's comprehension.

In terms of significance, the method shows considerable promise for application in image restoration, particularly for more complex and challenging inverse problems. The differentiation from non-blind methods is an important aspect, and the potential for this method to address a wider range of problems enhances its relevance and impact in the field.

**Weaknesses:**

The paper has several areas that require improvement. The theoretical analysis in section 3.1 is noted to need thorough revision, suggesting that some foundational aspects of the method may not be as robust as they could be. Additionally, much of the theoretical content is heavily dependent on related work, which might limit the perceived novelty of the contribution.

Specific areas needing attention include:
- The phrase "better neural network architecture choices" on lines 21-22 should be made more specific to avoid ambiguity.
- Equations 1, 3, 4, 6, 8: \DeclareMathOperator*{\argmin}{arg\,min}
- Line 129: If a certain coefficient $\rho$ is equivalent to Eq. 3, it should be easily expressible in terms of $\lambda$.
- Line 130: $g_*$ is not defined.
- Lines 132-134: The wording and notation need clarification (e.g., $\| z \|^2 = N_x \times M_x$, usually $\| z \|^2$ is a scalar, $N_x \times M_x$ is a tuple)
- The derivation of Eq. 8 seems out of place and might be better suited for section 3.2.3.

The clarity of the introduction could be improved as it currently reads too much like related work. The work of Chung et al. [2, 3] should be discussed separately in the related work section. Additionally, the statement about GAN inversion methods on lines 104-105 should consider other methods that optimize intermediate latent variables (e.g., https://arxiv.org/abs/1703.03208, https://arxiv.org/abs/2102.07364).

Algorithm 1 is confusing with its indices, and it is unclear whether $x_T$ or $x_0$ is the initial latent variable. Similarly, Algorithm 2 would benefit from including requirements similar to those in Algorithm 1 to enhance clarity.

Finally, while the results are significant, the paper would benefit from highlighting the unique theoretical contributions more clearly. The differentiation from non-blind methods, especially in solving more general or harder inverse problems, should be emphasized to better showcase the method's potential and impact.

In summary, while the paper presents a novel and promising approach to image restoration using pretrained diffusion models, it would benefit from addressing the theoretical and clarity issues mentioned above. Strengthening these aspects will enhance the overall quality and impact of the work.

**Questions:**

How does the method work for general inverse problems? Does Algorithm 1 actually require the measurements to be in image space?

**Limitations:**

- The method is presented as superior to non-blind methods [2, 3], but these methods tackle more general inverse problems that do not require measurements in image space (e.g. sparse-view CT or phase retrieval). This difference should be emphasized.
- It would be beneficial to know which problems cause the method to fail, at least in supplementary materials.

---

> ### Author Rebuttal · Authors · 2024-08-05
>
> Thank you for your feedback. Please see the detailed response below.
>
> ### Q2. Specific areas needing attention
>
> >The phrase "better neural network architecture choices" on lines 21-22 should be made more specific to avoid ambiguity.
>
>
> thanks. We mean that better NN architectures (for example, Transformers) have led to better generative models. We will make this clearer in our revision.
>
>
> >Equations 1, 3, 4, 6, and 8: \DeclareMathOperator*{\argmin}{arg,min}
>
> thanks for pointing this out. We will correct all of them in our revision.
>
> > Lines 132-134: The wording and notation need clarification
>
> thanks. We will make the notation more clear.
>
> >The derivation of Eq. 8 seems out of place and might be better suited for section 3.2.3.
>
> thanks. We will move it to section 3.2.3.
>
>
> ### Q3. introduction enhancement and [2, 3] should be discussed separately
>
> thanks for pointing this out. We will adjust the introduction and discuss [2, 3] in the related work.
>
> ### Q4. Statement about GAN inversion methods on lines 104-105
>
> Thanks for pointing this out. We will include the mentioned works in our revision.
>
> ### Q5. Algorithm 1 and 2 are confusing with their indices.
>
> Thanks for pointing this out. We will use clear indices for both Algorithm 1 and Algorithm 2 in our revision.
>
>
> ### Q6. Solving more general IR tasks?
>
> BIRD can be applied to a wide range of IR tasks, not just the blind ones. BIRD can solve non-blind tasks (known degradation operators) like inpainting and colorization. We refer the reviewer to Figure 1 in our rebuttal PDF, where we show some visual results of our method solving more IR tasks. Our method remains the same as for the blind case, except that we do not optimize for the degradation operator (as it is known).
>
> ### Q7. Solving IR tasks where the measurements are not in image space?
>
> Yes, BIRD can handle IR tasks where the measurements are not in image space. Our method only requires a differentiable degradation operator (or some differentiable approximation of it). BIRD can handle the non-differentiable JPEG-deartifacting. In Figure 1, we show some visual results of the sparse-view CT.
>
> ### Q8. The differentiation from non-blind methods, BIRD's potential and impact
>
> Although BIRD is primarily proposed as a blind zero-shot method, here we discuss some of the similarities/differences with non-blind methods. To further showcase the potential of BIRD, we highlight three aspects that demonstrate the advantages of our method.
>
> *Robustness to severe degradation*
>
> As depicted in Figure 2 of our rebuttal PDF, BIRD is more robust to severe degradation (SRx16), both in terms of image quality and faithfulness.
>
> *Robustness to noise distribution*
>
> Real noise is usually not Gaussian. We compare BIRD with non-blind methods using a mixture of Gaussian and speckle noise.  As shown in Figure 3 of our rebuttal PDF, although DPS[2] is  robust to noise distribution, BIRD generates higher quality images.
>
> *Robustness to error in the degradation model*
>
> Non-blind methods generally rely on an off-the-shelf method to estimate the degradation operator. These off-the-shelf methods are not 100% accurate. Thus, a valuable feature of a non-blind IR method is its robustness to errors in degradation operator estimation. In Figure 4 of our rebuttal PDF, we show some visual results where we simulate a small error in the degradation operator. BIRD produces better quality outputs.
>
>
> |   **Method**|   **SR x16**| **SRx4  (noise with a mixture model)** |**Deblur (degradation operator with a small error)** |
> |---|---|---|---|
> |  DDNM[20] |  21.73/0.451 | 21.26/0.472  | n.a  |
> |  DPS[2] |  21.13/0.447 | 23.76/0.277  | 23.86 /0.360  |
> |  BIRD | 21.95/0.349  | 24.82/0.239  |  24.56/0.251  |
>
> Quantitative Comparison (PSNR/LPIPS) with state-of-the-art non-blind methods.
>
> ### Q9. Problems causing the method to fail
>
> BIRD needs a parametric form of the degradation model (albeit with unknown parameters). Such a constraint is not satisfied for some IR problems like image deraining, and so our method cannot handle it. We mention this limitation in our main paper at lines (233-236).

---

> ### Author Response · Authors · 2024-08-08
>
> We are sorry, we did not manage to include this part during our initial version of rebuttal.
>
> ### Q1. The theoretical analysis in Section 3.1 needs revision
>
> Thanks for pointing this out. We will update Section 3.1 and take the comments of the reviewer about "specific areas needing attention" into account, including fixing the notation issues.
>
> Here, we provide a brief derivation that leads to the same final result and comment on the theory of BIRD. (We keep the same notation as in the paper.)
>
> We begin with Eq. (3):
> $$
> \hat{x} = \arg\min_{x \in \mathbb{R}^{N_x \times M_x}} \| y - H(x) \|^2 + \lambda R(x)
> $$
>
> $R$ is a prior term, and $\lambda > 0$ trades off the likelihood and the prior.
>
> We define $\Omega \subset [0, 255]^{N_x \times M_x}$ as the domain of "realistic" images (i.e., the support of $p(x)$) and propose employing a formulation that implicitly assumes a uniform prior $p_U$ on $\Omega$. That is, $R(x) = - \log(p_U(x)) = -\log(\text{const} \cdot 1_\Omega(x))$, where $1_\Omega(x)$ is 1 if $x$ is in the support $\Omega$ and 0 otherwise.
>
>
>  This results in the following formulation:
>
> $$
> \hat{x} = \arg\min_{x \in \Omega} \| y - H(x) \|^2
> $$
>
> In theory, this choice ensures two key aspects:
>
> 1. *Guarantees realism:* By definition, we are restricting the search to $\Omega$. This is particularly illustrated in Figure 4 of our rebuttal, where BIRD generates more realistic results even in the presence of an error in the degradation model.
>
> 2. *Favors higher fidelity:* In other plug-and-play methods that use $R(x) = -\log(p(x))$, an image with higher $p(x)$ but lower fidelity (data term) can be favored over an image with a higher fidelity but a lower $p(x)$ (e.g., in the tail of the distribution). In contrast, in our case, the image with higher fidelity is always favored. This is particularly illustrated in Figure 2 of our rebuttal, where BIRD generates results with higher fidelity even under severe degradation.
>
> Returning to the derivation, to ensure that $x \in \Omega$, we parameterize $x$ via the initial noise of a pre-trained diffusion model $g$:
>
> $$
> \hat{x} = \arg\min_{z \sim \mathcal{N}(0, I)} \| y - H(g(z)) \|^2
> $$
>
> Given that most of the density of a high-dimensional normal random variable is around $\|z\|^2 = N_x M_x$ [12, 19], we can derive Eq. (8) (shown in the paper).
>
>
> We will be happy to address any further concerns the reviewer may have.
>
> ### Q9. Problems causing the method to fail
>
> For the non-blind case, as suggested by the reviewer, we experimented with the task of phase retrieval and found that there were occasional convergence issues. A similar problem was reported in [2]. We will mention it in the limitations section.

---

> > ### Comment · Reviewer_ebAr · 2024-08-09
> >
> > I appreciate the authors detailed responses to my and the other reviewers' comments.
> > The planned revisions and additional clarifications enhance the paper's robustness and clarity. I believe these changes will improve the paper's quality and impact.
> >
> > > Yes, BIRD can handle IR tasks where the measurements are not in image space. Our method only requires a differentiable degradation operator (or some differentiable approximation of it). BIRD can handle the non-differentiable JPEG-deartifacting. In Figure 1, we show some visual results of the sparse-view CT.
> >
> > In Figure 1, in the case of sparse-view CT, the term "degraded" might be misleading. The measurements in this scenario are not in image space; they should be sinograms instead. Are you perhaps referring to baseline reconstructions derived from these measurements as "degraded"?
> > If so, this interpretation differs from the conventional understanding of degradation used in inpainting tasks, where the degradation typically occurs within the image space itself.
> > Also missing here is the extent to which the "sparse view" was applied.
> >
> > My initial concern regarding the reliance on existing theoretical content, which could diminish the perceived novelty of the contribution, remains unaddressed. While I appreciate the revisions made to Section 3.1, the emphasis on problem formulation appears to be somewhat minor and does not directly contribute to the core approach discussed in Section 3.2.3. A more focused theoretical analysis explaining the significance and utility of equations (19), (21), and (22) would be far more beneficial and would strengthen the overall argument of the paper.

---

> ### Author Response · Authors · 2024-08-12
>
> Sorry for the delay. We would like to thank the reviewer for the fruitful discussion and the valuable comments that have helped improve our paper. Please find our detailed response below.
>
> ### Q10. Sparse-View CT in Figure 1
>
> Thank you for pointing this out. We agree that the term "degraded" may not be the most accurate description. For the sparse-view CT, we adopted the same setting as [3] and used their official implementation available on GitHub (specifically, the "run\_CT\_recon.py" file) to generate the measurements, which are not in the image space. The sparsity level is set to 6. For an input tensor of size [1, 1, 256, 256], the sinogram has a size of [1, 1, 363, 30]. In Figure 1, we show the baseline reconstructions derived from these measurements (similar to Figure 4 in [3]).
>
> ### Q11. Section 3.2.3, Prior Work, Theoretical Analysis, Eq (19), Eq (21), Eq (22) [Part 1]
>
> Here, we provide a more theoretical analysis of the approach discussed in Section 3.2.3 and comment on Eq (19), Eq (21), and Eq (22).
>
> First, we want to note that the derivation in Section 3.2.3 is directly connected to our problem formulation as we aim to solve  Eq (8):
>
> $$
> \hat{x} = \arg\min_{z: \|z\|^2 = N_x M_x} \| y - H(g(z)) \|^2
> $$
>
> One challenge in solving Eq (8) is that an iterative approach is computationally expensive, as a single evaluation of $g(z)$ may take minutes.
>
> To address this, we introduce a family of generative processes $\text{DDIMReverse}(., \delta t) = g^{\delta t}$, parameterized by $\delta t \geq 1$. These processes are defined *not on all latent variables* $x_{1:T}$, but on a subset \{$x_0, x_{\delta t}, x_{2 \delta t}, \ldots, x_{T-2 \delta t}, x_{T-\delta t}, x_T$ \} of length $K$. The scalar $\delta t$ defines the jump in the generative process sampling.
>
> We can show that, by carefully factorizing both the diffusion forward process and the corresponding generative process and choosing the appropriate marginals, for all $\delta t$, $g^{\delta t}$ constitutes a valid generative model from $p(x)$.
>
> Here is a brief derivation:
>
> We consider the following factorization of the diffusion forward process:
>
> $$
> q_{\delta t}(x_{1:T} | x_0) = q_{\delta t}(x_T| x_0)  \prod_{i=1}^{K} q_{\delta t}(x_{(i -1) \delta t} | x_{i \delta t}, x_0) \prod_{t \notin  \\{ 0, {\delta t}, {2 \delta t}, \ldots, {T-2 \delta t}, {T-\delta t}, T  \\} } q_{\delta t}(x_t | x_0)
> $$
>
> where
> $
> q_{\delta t}(x_t | x_0) = \mathcal{N} \left( \sqrt{\bar{\alpha_t}} x_0, (1 - \bar{\alpha_t}) I \right)
> $
> and
> $
> q_{\delta t}(x_{(i -1) \delta t} | x_{i \delta t}, x_0) = \mathcal{N} \left( \sqrt{\bar{\alpha_{(i -1) \delta t}}} x_0 + \sqrt{1 - \bar{\alpha_{(i -1) \delta t}} - \sigma_{i \delta t}^2} \frac{x_{i \delta t} - \sqrt{\bar{\alpha_{i \delta t} }} x_0}{\sqrt{1 - \bar{\alpha_{i \delta t} }}}, \sigma_{i \delta t} ^2 I \right)
> $
>
> Specifically, $q_{\delta t}(x_{(i -1) \delta t} | x_{i \delta t}, x_0) $ is defined such that $q_{\delta t}(x_{i \delta t} | x_0) = \mathcal{N} \left( \sqrt{\bar{\alpha_{i \delta t}}} x_0, (1 - \bar{\alpha_{i \delta t}}) I \right)$ for all $i \in [1, K]$. (Here, all the marginals $q_{\delta t}(x_t | x_0)$ for $t \in [1, T]$ match those in the original diffusion formulation.)
>
> We then define the corresponding generative process $p_{\theta, \delta t}$:
>
> $$
> p_{\theta, \delta t}(x_{1:T} | x_0) = p_{\theta, \delta t}(x_T) \prod_{i=1}^{K} p_{\theta, \delta t}(x_{(i -1) \delta t} | x_{i \delta t}) \prod_{t \notin  \\{ 0, {\delta t}, {2 \delta t}, \ldots, {T-2 \delta t}, {T-\delta t}, T  \\}} p_{\theta, \delta t}(x_0 | x_t)
> $$
>
> (Only the first part of the factorization is used to produce samples.)
>
> We define
>
> $
> p_{\theta, \delta t}(x_0 | x_t) = \mathcal{N} \left( f_{\theta}^{(t)}(x_t), \sigma_t^2 I \right)
> $
> and
> $
> p_{\theta, \delta t}(x_{(i -1) \delta t} | x_{i \delta t}) = q_{\delta t}(x_{(i -1) \delta t} | x_{i \delta t}, f_{\theta}^{(i \delta t)}(x_{(i -1) \delta t}))
> $
> where
> $
> f_{\theta}^{(t)}(x_t) = \frac{x_t - \sqrt{1 - \bar{\alpha_t}} \cdot \epsilon_\theta^{(t)}(x_t)}{\sqrt{\bar{\alpha_t}}}
> $
>
> We optimize $\theta$ using the variational inference objective:
>
> $$
> J_{\delta t}(\epsilon_\theta) = \mathbb{E_{x_{0:T} \sim q_{\delta t} (x_{0:T})}} \left[ \log q_{\delta t} (x_{1:T} | x_0) - \log p_{\theta, \delta t} (x_{0:T}) \right]
> $$
>
> $$
> = \mathbb{E_{x_{0:T} \sim q_{\delta t} (x_{0:T})}} \left[ \sum_{i=1}^{K} D_{KL} \left(q_{\delta t} (x_{(i -1) \delta t} | x_{i \delta t}, x_0)  \| \|  p_{\theta, \delta t} (x_{(i -1) \delta t} | x_{i \delta t}) \right) + \sum_{t \notin  \\{ 0, {\delta t}, {2 \delta t}, \ldots, {T-2 \delta t}, {T-\delta t}, T  \\}} D_{KL} \left(q_{\delta t} (x_t | x_0)  \|\|   p_{\theta, \delta t} (x_0 | x_t) \right) \right]
> $$
>
> where each KL divergence is between two Gaussians where only the mean depends on $\theta$.

---

> ### Author Response · Authors · 2024-08-12
>
> ### Q11. Section 3.2.3, Prior Work, Theoretical Analysis, Eq (19), Eq (21), Eq (22) [Part 2]
>
> By substituting all the terms with their values:
>
> $$
> J(\epsilon_\theta) \equiv \mathbb{E_{x_{0:T} \sim q_{\delta t} (x_{0:T})}} \left[ \sum_{i=1}^{K} \frac{\| x_0 -  f_{\theta}^{(i \delta t)}(x_{i \delta t})  \|^2}{2 \sigma_{i \delta t}^2}   + \sum_{t \notin  [ 0, {\delta t}, {2 \delta t}, \ldots, {T-2 \delta t}, {T-\delta t}, T  ]} \frac{\| x_0 -  f_{\theta}^{(t)}(x_{t})\|^2}{2 \sigma_t^2}  \right]
> $$
>
> $$
> \equiv \mathbb{E_{x_{0:T} \sim q_{\delta t} (x_{0:T})}}  \sum_{t=1}^{T}  \frac{\| x_0 -  f_{\theta}^{(t)}(x_{t})\|^2}{2 \sigma_t^2}  \equiv \mathbb{E_{x_0 \sim q(x), \epsilon \sim \mathcal{N}(0, I), x_t = \sqrt{\bar{\alpha_t}} x_0 + \sqrt{1 - \bar{\alpha_t}} \epsilon}}  \sum_{t=1}^{T}  \frac{\| \frac{x_t - \sqrt{1 - \bar{\alpha_t}} \epsilon}{ \sqrt{\bar{\alpha_t}}} -  \frac{x_t - \sqrt{1 - \bar{\alpha_t}} \cdot \epsilon_\theta^{(t)}(x_t)}{\sqrt{\bar{\alpha_t}}})\|^2}{2 \sigma_t^2}
> $$
>
>
> $$
> J(\epsilon_\theta) \equiv \mathbb{E_{t \sim \mathcal{U}(0, 1); x_0 \sim q(x); \epsilon \sim \mathcal{N}(0, I)}} \left[ \| \epsilon - \epsilon_{\theta} (\sqrt{\bar{\alpha_t}} x_0 + \sqrt{1 - \bar{\alpha_t}} \epsilon, t) \|^2 \right]
> $$
>
> The objective $J_{\delta t}(\epsilon_\theta)$ matches the original training objective of the vanilla diffusion model (defined over all time steps $[1, T]$). This implies that:
>
> 1) All the generative processes $g^{\delta t}$ are equivalent and can be used to sample from $p(x)$.
> 2) There is no need for retraining to sample from $g^{\delta t}$, as the training objective for the generative process involving only a subset of the latent variables is the same as the one corresponding to the vanilla diffusion model (which involves all the latent variables).
>
> Based on these results, we can rewrite the minimization in eq (8) as follows:
>
> $$
> \hat{x} = \arg\min_{z: \|z\|^2 = N_x M_x} \| y - H(g^{\delta t \gg 1}(z)) \|^2
> $$
>
> This leads to a more efficient optimization due to the much faster evaluation of $g^{\delta t \gg 1}(z)$ (as the length of the sampling trajectory is much shorter than $T$), while remaining equivalent to eq (8).
>
> In eq (19), given an initial random Gaussian sample $z$, we use $g^{\delta t \gg 1}$ instead of the original $g$ to evaluate the optimization objective. Once $\| y - H(g^{\delta t \gg 1}(z)) \|^2$ is computed, we take a vanilla gradient descent step with respect to the optimization variables ($z$ and $\eta$), which leads directly to eq (21).
>
> However, when applying the gradient step, $z$ may deviate from the space \{ $z: \|z\|^2 = N_x M_x$ \}, so we project it back, which leads to eq (22).
>
> We repeat this process until convergence.
>
> Although that we build on existing techniques, our work introduces a novel formulation of inverse problems within the context of diffusion models and solves it in a unique and innovative way. We also would like to emphasize that we are the *first* to demonstrate that an inversion-based IR approach in the context of diffusion models is not only feasible but also highly effective.
>
> The reviewer aptly summarizes this by stating, *"The simplicity and novelty of the method lie in its creative combination of well-known techniques, making it a noteworthy contribution to the field."*
>
> As for the impact of our work, we believe that BIRD represents a significant leap forward in expanding the applicability of diffusion models to inverse imaging. BIRD provides a fresh perspective, and we hope it will inspire further research into the under-explored concept of inversion within the context of inverse problems and diffusion models.

---

### Official Review · Reviewer_PrQF · 2024-07-12

**Soundness:** 3
**Presentation:** 2
**Contribution:** 3
**Rating:** 5
**Confidence:** 4

**Summary:**

This paper proposed new blind image restoration (BIR) method by exploring the image prior induced by diffusion model (DM). Different from the existing DM-based methods, this work presents a diffusion inversion technique, such that the estimated image can be constrained to lie in the image manifold learned by the pre-trained DM, and the computational cost can be reduced. Experiments on several image restoration tasks have been conducted to demonstrate the effectiveness of the proposed method.

**Strengths:**

The idea of diffusion inversion is interesting and reasonable, which broaden the concept of deep generative prior, and can possibly motivated more related studies.

**Weaknesses:**

The major problem is that the experiments are not comprehensive enough.

- Some commonly used benchmarks in corresponding BIR tasks were not considered. For example, for the blind motion deblur task, commonly used Lai [1] and Levin [2] datasets were not used for evaluation.

- Some advanced deep prior-based methods for corresponding image restoration tasks were not compared. For example, for the deblur task, some DIP-based methods e.g., [3][4][5], were not compared; for the super-resolution task, also some DIP-based ones, e.g., [6][7], were not included

References:

[1] W. Lai et al. A comparative study for single image blind deblurring. CVPR, 2016.

[2] R. Köhler et al. Recording and playback of camera shake: Benchmarking blind deconvolution with a real-world database. ECCV, 2012.

[3] D. Ren et al. Neural blind deconvolution using deep priors. CVPR, 2020.

[4] D. Huo et al. Blind image deconvolution using variational deep image prior. TPAMI, 2023.

[5] J. Li et al. Self-supervised blind motion deblurring with deep expectation maximization. CVPR, 2023.

[6] J. Liang et al. Flow-based kernel prior with application to blind super-resolution. CVPR, 2021.

[7] Z. Yue et al. Blind image super-resolution with elaborate degradation modeling on noise and kernel. CVPR, 2022.

**Questions:**

It seems that Figure 1 indeed shows an example of deblurring with motion blur, while the caption says Gaussian blur.

**Limitations:**

The authors have discussed the limitations, which are reasonable.

---

> ### Author Rebuttal · Authors · 2024-08-05
>
> Thank you for your feedback. Please see the detailed response below.
>
> ### Q1. Add more comparisons [3-7]
>
>
> |   Method|   Zero-shot?| Task-agnostic?  |  Code available? |   |
> |---|---|---|---|---|
> | BIRD  | *Yes*  | *Yes*  |  N.A |   |
> |  [3] |  Yes | No  |  Yes |   |
> |  [4] | Yes  | No  | Yes  |   |
> | [5]  |  No | No  |  No* |   |
> | [6]  |  Yes | No  | Yes  |   |
> | [7]  | No  | No  |  Yes |   |
>
>
>
> We thank the reviewer for the suggestion. BIRD is zero-shot and task-agnostic, while [4-7] are dataset-based (involving training) and [3-7] are task-specific. We followed recent zero-shot works, such as [8, 9], which do not consider dataset-based approaches, as such a comparison is not apple-to-apple. Nonetheless, we report the mentioned comparison in the following tables and will include them in our revision. Unfortunately, the inference script for [5] is not publicly available. We have contacted the authors and will include the comparison in a comment if we receive the scripts.
>
> |   **Method**|   **CelebA**| **Imagenet** |   **Lai[1]** | **Levin[2]**|
> |---|---|---|---|---|
> |  [3] |  20.12 | 19.15  |  21.13 |  33.07  |
> |  [4] |  21.43 | 20.45  |  **25.12** |  -  |
> |  BIRD | **24.67**  | **23.76**  |  24.57 | **34.18**  |
>
> Quantitative comparison (PSNR) on Gaussian Deblurring
>
> |   **Method**|   **CelebA**| **Imagenet** |
> |---|---|---|
> |  [6] |  20.62 | 20.45  |
> |  [7] |  22.38 | **22.53**  |
> |  BIRD | **22.75**  | 22.15  |
>
> Quantitative comparison (PSNR) on Super-resolution
>
> ### Q2. Benchmark on Lai [1] and Levin [2] Datasets for Blind Deblurring
>
> Lai [1] and Levin [2] datasets are generally used for dataset-based [7] and task-specific methods like [3, 4, 5]. We followed the recent zero-shot and task-agnostic works (the same category as our paper) such as [8, 9], which benchmark on ImageNet and FFHQ/CelebA.
>
> In the table above, we added a comparison on Lai [1] and Levin [2] for Gaussian deblurring. For [3] and [4], we report the results shown in their papers. We note that [4] did not compare on Levin [2].
>
> ### Q3. Figure 1 shows an example of deblurring with motion blur and not Gaussian blur
>
> Thanks for pointing this out. We will correct it in our revision.
>
>
> [8] Denoising Diffusion Restoration Models. Neurips 2022
>
> [9] Zero-Shot Image Restoration Using Denoising Diffusion Null-Space Model. ICLR 2023

---

### Official Review · Reviewer_gDW8 · 2024-07-15

**Soundness:** 3
**Presentation:** 3
**Contribution:** 2
**Rating:** 7
**Confidence:** 3

**Summary:**

The paper presents a method to accelerate blind image reconstruction by leveraging pre-trained diffusion models. The authors suggest a strategy that simultaneously optimizes the degradation model parameters and the restored image, thereby improving the reconstruction process’s efficiency.

Furthermore, the authors introduce a sampling method based on a pre-trained diffusion model. This method is devised to ensure the restored images are in alignment with the image manifold, a critical factor in preserving the restored images’ integrity.

To enhance the speed of the process, the authors consider advancing in the forward diffusion process using large time steps, Thus they are able to significantly reduce the runtime of restoration process.

The experimental results demonstrate improved performance across various tasks, outperforming BlindDPS, a recent solution for blind inverse problems. Furthermore, it surpasses a couple of other blind image reconstruction methods that do not rely on diffusion models.

**Strengths:**

- The paper demonstrates a significant improvement in speed and memory efficiency over the primary method, BlindDPS. Additionally, the quality of the reconstruction has also been enhanced.

- The method, in particular, is articulated and presented effectively in the paper.

**Weaknesses:**

- The paper lacks a theoretical discussion and experimental comparison with some recent or related blind diffusion methods for inverse imaging problems, such as [3-5].

    - [3] “Fast Diffusion EM: A Diffusion Model for Blind Inverse Problems with Application to Deconvolution” by Laroche, Charles, Andrés Almansa, and Eva Coupete. *WACV* 2024.

    - [4] “Gibbsddrm: A Partially Collapsed Gibbs Sampler for Solving Blind Inverse Problems with Denoising Diffusion Restoration” by Murata, Naoki, et al. *ICML* 2023.

    - [5] “A Diffusion Model with State Estimation for Degradation-Blind Inverse Imaging” by Ji, Liya, et al. *AAAI* 2024.

**Questions:**

- It would be beneficial to include the results of the version with δt=1 in Tables 1, 3, and 4 as well.

- The paper only qualitatively considers JPEG de-artifacting and compares it solely to DIP. Why are more recent methods such as the following not considered?

    - [1] “Towards Flexible Blind JPEG Artifacts Removal” by Jiang, Jiaxi, Kai Zhang, and Radu Timofte. *ICCV* 2021.

    - [2] “DriftRec: Adapting Diffusion Models to Blind JPEG Restoration” by Welker, Simon, Henry N. Chapman, and Timo Gerkmann. *TIP* 2024.

- Minor point:

    - In Figure 3, the image index should be corrected from (f) to (e).

**Limitations:**

no comments.

---

> ### Author Rebuttal · Authors · 2024-08-05
>
> Thank you for your feedback. Please see the detailed response below.
>
>
> ### Q1. Theoretical Discussion and Experimental Comparison to [3-5]
>
>
> |   Method|   Zero-shot?| Task-agnostic?  |  Code available? |   |
> |---|---|---|---|---|
> | BIRD  | *Yes*  | *Yes*  |  N.A |   |
> |  [1] |  No | No  |  Yes |   |
> |  [2] | No  | No  | Yes  |   |
> | [3]  |  Yes | No  |  Yes |   |
> | [4]  |  Yes | Yes  | Yes  |   |
> | [5]  | No  | Yes  |  No |   |
>
>
>
> We thank the reviewer for the suggestion. We will add the content below to the paper. First, let us discuss the references [3-5]. BIRD is a zero-shot method (no training involved), while [5] involves fine-tuning a diffusion model and training a state estimator. It also requires a dataset of input-output pairs for training, whereas BIRD does not. References [3-4] are zero-shot approaches, but  fundamentally different from BIRD. A key difference is that [3-4] *modify* the original diffusion sampling by adding a projection-based step to enforce consistency with the corrupted image. In contrast, in BIRD we use the original diffusion sampling scheme as is. Moreover, [3] applies the EM algorithm after each diffusion reverse step to jointly update the image latent and the kernel blur. [4] extends DDRM to the blind case by adopting a Gibbs sampler to enable efficient sampling from the posterior distribution. We note that [3] is task-specific (proposed only for blind deconvolution) while BIRD can handle different IR tasks. [4] can only handle linear IR tasks (BIRD can handle non-linear ones like JPEG de-artifacting) and is not easily applicable to problems involving linear operators for which the SVD is computationally infeasible.
> Below we also show a quantitative comparison between BIRD and methods [3,4]. Unfortunately, we could not compare to [5] because we could not find code for it and the results in [5] is not directly comparable to our paper because of not using the exact degradation or noise model.
>
>
> |   **Method**|   Motion Deblur | Gaussian Deblur |   SR x8 |
> |---|---|---|---|
> |  [3] |  23.18/0.284 | 24.52/0.235  |  n.a |
> |  [4] |  22.94/0.314 | 23.57/0.266  |  22.17/0.357 |
> |  BIRD | **23.76**/**0.263**  | **24.67**/**0.225**  |  **22.75**/**0.306** |
>
> Quantitative comparison (PSNR/LPIPS) on CelebA.
>
>
>
> ### Q2. Comparison to [1-2]
>
> We thank the reviewer for their suggestion. We did not include comparisons to methods [1-2] because BIRD is zero-shot and task-agnostic, while [1-2] are dataset-based (involving training) and task-specific (JPEG deartifacting). In our paper, we followed recent zero-shot works, such as [8, 9], which do not consider dataset-based approaches, as such a comparison is not an apple-to-apple.
>
> Nonetheless, we compared to [1], which provides a pretrained model in their official code. Unfortunately, we could not compare to [2] as they do not provide a pretrained model and we could not train their model in time. We will include a comparison to [2] in our revision.
>
> In the table below, we illustrate the key difference between zero-shot (single-image) and dataset-based methods. Dataset-based approaches tend to work well when tested with data having the same degradation operators/noise distributions seen during training, but their performance drops when dealing with new data. We further show this key difference in Figure 5 of the rebuttal PDF. In that Figure 5, we show that when testing [1] with no noise (the same as during training), it outputs a clean image. However, when adding a small amount of noise not seen during training, [1] produces artifacts. In contrast, BIRD yields the same result under both settings.
>
> |   **Method**|   **CelebA (w/o noise)**| **CelebA (w/ noise)** |
> |---|---|---|
> |  [1] |  **29.24**/0.242 | 26.45/0.318  |
> |  BIRD | 28.75/**0.221**  | **28.67**/**0.224**  |
>
> Quantitative comparison (PSNR/LPIPS) on JPEG de-artifacting on CelebA
>
> ### Q3. Include the Results of the Version with δt=1 in Tables 1, 3, and 4
>
> We show below the updated Table 3. Running BIRD with δt=1 is computationally almost infeasible as one image take around 22500 seconds. Indeed, one of our motivations is to propose a fast (and computationally feasible) diffusion inversion that benefits from the ability to jump ahead in the diffusion reverse process (δt higher than 1).
> We will add the results for some reasonably small δt (δt=10 or δt=20)  for other tables in our revision.
>
> (If the reviewer means the case with one jump (δt=1000), this is so feasible and we can add it.)
>
> |   step size (δt) |   PSNR ↑  | LPIPS ↓  |  Time [s] ↓ |
> |---|---|---|---|
> |  δt=1 |  - | -  |  22500 |
> |  δt=50 | 28.74  | 0.218  | 412  |
> | δt= 100  |  28.67 | 0.224  |  234 |
> | δt= 200  |  28.45 | 0.237  | 110  |
>
> Updated Table 3
>
> ### Q4. In Figure 3, the image index should be corrected from (f) to (e)
>
> Thanks for pointing this out. We will correct it in our revision.
>
>
> [8] Denoising Diffusion Restoration Models. Neurips 2022
>
> [9] Zero-Shot Image Restoration Using Denoising Diffusion Null-Space Model. ICLR 2023

---

> > ### Comment · Reviewer_gDW8 · 2024-08-14
> >
> > I appreciate the authors’ efforts in addressing many of the reviewers’ comments. I am inclined to remain positive on the score, as most of my original concerns have been addressed. However, I look forward to seeing the feedback from a couple of other reviewers before deciding on their remaining concerns.

---

### Author Rebuttal · Authors · 2024-08-07

We would like to thank all the reviewers for their time and feedback. We attached a pdf containing some additional results based on their comments and suggestions.

---

### Comment · Area_Chair_ENah · 2024-08-13
**TLDR: Reviewers please do acknowledge the rebuttals and react to them.**

Dear reviewers,

thanks for your reviews.  Please do look at the rebuttals of the author and give the authors some feedback whether they could address your concerns.  This is really important for the authors.

Best regards Your AC.

---

### Decision · Program_Chairs · 2024-09-25

**Decision:**

Accept (poster)

**Comment:**

Scores 4 4 5 7

The paper proposes a new approach to blind image restoration using diffusion models.  Reviewer cmXb points out that the work is too close to existing methods, like GDP (Generative diffusion prior for unified image restoration and enhancement) and TAO (Test-Time Degradation Adaptation for Open-Set Image Restoration).  The rebuttal does clarify this.  Reviewer ebAr points out several problems in the presentation which the rebuttal again is able to address.  They also do provide some further theoretical analysis.  The concerns of reviewer PrQF were also reasonable addressed.  Overall the paper appears to be a solid contribution and can show with several experiments that their approach is better than other established reconstruction methods.

The average score of the paper is 5.0.  However, the lower scoring reviewers didn't increase their score, after their concerns were (mostly) addressed.  In summary, I suggest to accept this paper (if space permits).